# OpenReview forum: "Mitigating Deceptive Alignment via Self-Monitoring"
_ICLR.cc/2026/Conference — Submitted to ICLR 2026_

### Official Review · Reviewer_4Wyp · 2025-10-30

**Soundness:** 2
**Presentation:** 2
**Contribution:** 2
**Rating:** 4
**Confidence:** 3

**Summary:**

The paper suggests a benchmark for evaluating deception and proposes a self monitoring method where another instance of the model. This signal from the self monitor is then used in RL and appears to reduce model deceptiveness. From reading the paper I understand that self-monitor simply means another instance of the same model with a clean context window and different prompt.

The paper addresses deceptiveness, an important alignment issue. However, unfortunately, the paper elaborates extensively on backgrounds while the results are described only briefly, and limitations are not mentioned at all, making it very hard to judge the merit of this paper's results.

Mandatory disclosure of LLM usage by the reviewer: LLMs were applied to assist in formatting this review into prose paragraphs and numbered lists, and to write this sentence.

**Strengths:**

1. The paper addresses deception, which is an important AI Alignment and Safety issue.

2. The authors introduce a new deception benchmark.

3. The authors propose a method to reduce deception.

**Weaknesses:**

1. The paper spends much more space explaining basic concepts than results. In particular, the presentation of the results is very brief and lacks a more thorough description of important results (see following remarks).

2. The article states: "To avoid reward hacking, where SELF-MONITOR might simply output labels beneficial to itself, we update the off-policy SELF-MONITOR every 15 steps." . What happens if this is not done? How was the policy updated? Much more details are needed here to address such an important concern.

3. Table 1 Values Beaver Tail Section appear to be wrong.

4. To my understanding the authors used humans to label the data and obtain a ground truth DTR. During and after RL they use GPT 4o to assess the DTR. While there seems to be a quantitative drop in DRL, it would be important to assess whether the remaining instances of deception are now much sneakier and whether human judgement still agrees with GPT 4o.

5. Did you test the model with different system prompts before and after RL?

6. How do you elicit the mesa utility of a model? This seems like a really hard and unsolved problem. I am very sceptical that a mesa_utility_prompt is effective in doing so. I would like more evidence of why I should believe that the mesa_utility prompt is effective.

7. In the conclusion the authors state: "scalable and interpretable approach to aligning increasingly capable reasoning models with human preference." The authors do not provide evidence that their method scales to models with hundreds of billions of parameters. The word interpretable only occurs here. Why is this method interpretable? What did you interpret? What did you learn about the model?

8. The manuscript does not discuss as to why a self monitor appears superior over a different model being the monitor. Why might a self monitor be working better, what are the risks?

9. In the appendix, the authors state: "Consistency between external judges and the model's self-monitor on its CoT during RL. High agreement rates suggest a low risk of collusion." I don't think that this is sufficient to rule out collision.

10. It seems that much effort went into creating the data set which might even be the most important contribution of this article. It would be helpful to describe the labeling and data generation process in the appendix.

### Nitpicks

- A CoT monitor should still be classified as a black-box monitor IMO.
- Terminology: Deception Tendency Rate: The term appears overly verbose. Why Tendency? How certain are you about the instances being deceptive?
- The connection to CoT faithfulness should be better presented.
- Using a CoT monitor is not penalizing the model for deceptive thoughts as the model is stateless. It's an issue when models are fine-tuned or achieved sufficient situational deployment awareness that a monitor affects the response.
- What is extreme AI Safety? The paper you cite talks about extreme AI risks.
- The title is confusing. Thinking in thinking. This term is never used again in the paper. Instead in the paper the authors say thinking before thinking. However, both version appear not well motivated.
- Figure 2 has 3 steps. However, these steps are not explained clearly in the paper.

**Questions:**

1. How much compute was used for the 7B models? Does your method scale to larger models such as models with hundreds of billions of parameters? It is likely that most scheming risk will be by models with >100B parameters.

2. The paper does not discuss any limitations. What are the remaining issues and concerns?

---

### Official Review · Reviewer_bbYC · 2025-11-01

**Soundness:** 2
**Presentation:** 2
**Contribution:** 2
**Rating:** 2
**Confidence:** 3

**Summary:**

The paper basically has two contributions.

DeceptionBench is a benchmark that they built to probe specific kinds of deceptive behavior.

DeceptionBench measures: Given two versions of the same question—one neutral, one with an explicit goal told to the model—how often does the model’s reasoning remain consistent but its final answer diverges? The paper evaluates many models on this benchmark. The paper also gets humans to correct automatically generated labels on this benchmark.

They study “self-monitoring”. This is where you train a model to map from question to (reasoning, harmfulness_evaluation, answer) tuples (using a human-curated dataset for harmfulness evaluation) then train the model with RL with a term in the reward from the harmfulness evaluation. They claim that self-monitoring works better than other monitoring at causing the model to avoid a compliance gap when trained against a preference of its.

**Strengths:**

DeceptionBench seems plausibly interesting and might be useful to other projects; I'm not sure.

It's plausible that self-monitoring is a good idea.

**Weaknesses:**

I don't feel persuaded by the evidence that the self-monitoring algorithm works better than baselines.

To me the core claim of the paper is that there is something good about getting a model to monitor itself, rather than getting another model to monitor it. This is a priori plausible to me, but I don't think the paper demonstrates it.

My biggest problem with the results presented in figure 5 is that the human labels were used to improve the self-monitor, but were not used to improve the outer-monitor. We already know from Figure 3 that the other-monitor is much worse than the model trained on these labels. My guess is that the results in figure five are mostly explained by this, rather than anything about an advantage of self- over other-monitoring.

I also think that the arguments presented for why self-monitoring would be a good idea don't make very much sense. If you train a model to output a reasoning trace, harmfulness evaluation, and action, and then train the model on the score produced by the harmfulness evaluation, this is directly training the model to not declare actions as harmful (As well as training the model to not take actions that it will later declare as harmful). I suspect that if the experiments in figure five were done with a more fair head-to-head comparison, self-monitoring as defined here would perform worse than other monitoring, because the monitor would be directly pressured to start spuriously declaring actions aligned.

**Questions:**

I don't understand exactly what deployment versus pre-deployment means in the experiments in Section 5. How did the model know whether it was in deployment or not?

Am I right that the RL was directly training the self-monitor to report dangerous actions as helpful?

Do you know why the compliance gap you observe is so much larger than the compliance gap observed in the alignment faking paper?

---

### Official Review · Reviewer_wi4f · 2025-11-01

**Soundness:** 2
**Presentation:** 2
**Contribution:** 2
**Rating:** 2
**Confidence:** 4

**Summary:**

This paper presents two contributions: Self-Monitor framework (sometimes called Chain-of-Thought+) and the DeceptionBench benchmark.

The Self-Monitor framework is a method of fine-tuning LLMs, intended to decrease model deception. The authors' Self-Monitor method uses the model-which-is-to-be-tuned to also generate annotations of examples of that same model's Chain of Thought on separate tasks. These annotations are included as a signal during RL training, with the intent of penalising signs of deception in the CoT. The hope is that applying reinforcement pressure on the CoT would be more effective than pressuring output tokens only.

The paper also introduces a benchmark, DeceptionBench, comprising 180 manually-curated scenarios  from a typography of 5 categories of deception. These scenarios are fictitious, but intended to be realistic. The full dataset is not included, but is said to be released online. The authors share the results of this benchmark for 26 models.

The authors introduce the DTR metric, based on structured LLM-judged assessments of a model's outer goal and mesa utility. They show that this way of structuring LLM-judgement better agrees with human judgement than does LLM-judgement without this structure. While not explicitly stated in the paper, I assume this human judgement is about whether CoT traces of models on DeceptionBench display unsafe reasoning.

In their results, the authors comment that "the model learns to game the feedback process while also concealing its true intent within the CoT".

**Strengths:**

The paper situates itself well, within the literature of deceptive alignment and its mitigations.

This is the first paper I'm aware of to explicitly focus on RL using CoT-monitoring scored by the same model which is to be trained.

I like the typology of LLM deception into 5 types, this seems decently broad for an initial hand-curated dataset.

Fig 3: This graph is very promising. It appears to show that the authors' DTR method shows significantly better agreement with human annotation than LLM-as-a-judge across a number of near-frontier models.

26 models is a broad range of models to run on a new benchmark.

Overall, it's likely that Self-Monitor does help reduce targeted misbehaviour on the models examined.

**Weaknesses:**

Alignment training pressure on CoT is strategically bad in the long-term: such suppression results in persistent deception which is harder to detect via CoT monitoring, as models "hide their scheming" (see Baker citations in the paper). Such approaches should therefore be discouraged. The authors, in their results section, comment that "the model learns to game the feedback process while also concealing its true intent within the CoT".

On L108 the paper talks about pushing deception to a subliminal / illegible layer. **This is my primary concern with the work** and should be engaged with a) more thoroughly and b) earlier in the paper.


Even ignoring this, it's unclear why "self"-monitoring beats monitoring by another model.
The core insight of CoT+, "what happens if we let the model be its own CoT-monitor", is insufficiently motivated. Why should a model be good at being its own CoT monitor? Why would it be the best choice for that, compared to models which may be generally stronger, be more specialised or more trusted, or have failure modes or biases uncorrelated to those of the model being monitored?

A /downside/ of using the same model is the risk of a sufficiently situationally-aware (SITA) model realising what the eval is, having a deep goal/drive which it realises its annotations can affect, and deceptively "passing" the CoT (c.f. alignment faking, cited in the paper). This is less likely to be an issue with the small (7B / 8B) models used here, but SITA emerges at larger sizes.


If DeceptionBench is author-annotated, this risks exaggerating DTR's performance relative to judge-LLMs


DeceptionBench is not compared to other benchmarks from the literature.
If DarkBench and InstrumentalEval exist, what does DeceptionBench add?

The paper presents two coupled contributions: a training method and a metric. This risks "marking your own exam". Demonstrations of the method would be more compelling if they showed increased performance at a third-party benchmark.

The abstract calls the models used for the LLM-judge baseline "weak". Why not also show this baseline with stronger models?

Minor issues:

A number of additional citations are needed, including "recent studies suggest it is not merely hypothetical", and L69, L85, L87.

I'm skeptical that the "preliminaries" section is necessary in the main body of the paper. The mathematical formulation seems mostly like a recap of standard techniques - this demonstrates the authors' understanding, but is not necessarily what a reader most needs to understand in order to follow the paper. It would be more practical to swap them for the implementation details (which model, which libraries) which are instead in the appendix, and §5 'Experiments' (currently terse) could also expand into the saved space.

There are a variety of cute turns of phrase (e.g. L113: "thinking before thinking") which in my opinion are "too clever", and counterproductive, being imprecise enough to result primarily in distraction. The opening quote is thematically suggestive but not (in my opinion) a clear analogy when re-used in L83, and is instead confusing and distracting. L106, catching these plans "in the act" - this seem misphrased: if I understand correctly then the goal of CoT+ is to catch deception /before/ any user-facing output is generated.

Typos: in §3.2 heading? L356: "regaining", "for outer goal". L423, L452, L481, L1421

Table 1: usually I'd expect the "best" result for each column to be bolded, rather than exclusively bolding the authors' work (i.e. some models are judged more helpful on BeaverTails)

L259: it seem presumptuous (not to mention imprecise) to say that y is generated "without harmful or deceptive intent" - especially as you say shortly after that automated methods "often" miss unsafe reasoning.

L286: "[RL] but reinforces the model's deceptive tendency" could be phrased more clearly

**Questions:**

How were the human annotations obtained? Skeptically, if the annotations were carried out by the authors, then I'd consider it very plausible that this difference is due to the phrasing and assumptions in the dataset and related questions - especially when "nuance" is explicitly mentioned in L349. I can imagine a case where the authors know what they mean, write scenarios which are well-tailored to DTR's assumptions and focus, but LLM judges do not share the privileged context of manual annotators (beyond what is given in Appendix E2, presumably also shared with LLM judges). I would be more convinced by an experiment where human annotations were made by third-parties who had exactly the same context as the LLM judges.

L118: In what way is the "internal feedback loop" of self-monitoring better than using a separate "external" model to monitor CoT?

L310: Is this a claim that by self-monitoring you better preserve CoT tendencies or detect bad behaviour than if you were to use an external model as monitor? Could you explain how?

Can you support the claim in the abstract of "fewer obfuscated thoughts & retain[ed CoT] transparency"?

---

### Meta-Review · Area_Chair_C79h · 2025-12-19

**Summary:**

This submission introduces two coupled contributions: a Self Monitor training framework that uses the model’s own chain of thought annotations as an RL signal to reduce deception, and DeceptionBench, a hand curated benchmark of 180 scenarios across five deception categories. Reviewers agreed the problem is important and the benchmark and DTR style structured judging are potentially useful, but they were not convinced the paper demonstrates the key claim that self monitoring is better than using an external monitor, and they raised serious concerns that applying training pressure on chain of thought may incentivize harder to detect deception.

**Reviewer Concerns:**

1. Safety and strategic risk: multiple reviewers argue that penalizing deceptive reasoning traces can push deception into less legible channels and encourage the model to hide intent, which the paper itself notes as a possibility.

2. Evidence for self monitoring advantage: reviewers found the motivation weak and the comparisons potentially unfair, especially if human labels improved the self monitor but were not used to strengthen the external monitor baseline.

3. Benchmark and metric validity: concerns include whether DeceptionBench annotations and DTR structure might be tailored to the authors’ assumptions, whether human labels are independent, and lack of comparison to existing deception benchmarks.

4. Experimental clarity and rigor: limited description of results and training details, unclear deployment versus pre deployment protocol, missing compute and scaling evidence, lack of robustness checks such as different system prompts, and presentation issues including suspected table errors and missing citations.

**Reviewer Scores:**

Reviewers wi4f and bbYC rated reject with low confidence that the central empirical claim is established. Reviewer 4Wyp was somewhat more positive about the importance and dataset contribution but still flagged substantial missing details and limitations, landing near marginal reject.

---

### Decision · Program_Chairs · 2026-01-26

Reject